# Effects of Different Lengths of Oligo (Ethylene Glycol) Side Chains on the Electrochromic and Photovoltaic Properties of Benzothiadiazole-Based Donor-Acceptor Conjugated Polymers

**DOI:** 10.3390/molecules28052056

**Published:** 2023-02-22

**Authors:** Songrui Jia, Shiying Qi, Zhen Xing, Shiyi Li, Qilin Wang, Zheng Chen

**Affiliations:** 1Key of High Performance Plastics, Ministry of Education, National & Local Joint Engineering Laboratory for Synthesis Technology of High Performance Polymer, College of Chemistry, Jilin University, Changchun 130012, China; 2Department of Materials Science, Fudan University, Shanghai 200433, China

**Keywords:** oligo (ethylene glycol) (OEG), electrochromic, photovoltaic, benzodithiophene (BDT)

## Abstract

In recent years, donor-acceptor (D-A)-type conjugated polymers have been widely used in the field of organic solar cells (OSCs) and electrochromism (EC). Considering the poor solubility of D-A conjugated polymers, the solvents used in material processing and related device preparation are mostly toxic halogenated solvents, which have become the biggest obstacle to the future commercial process of the OSC and EC field. Herein, we designed and synthesized three novel D-A conjugated polymers, PBDT1-DTBF, PBDT2-DTBF, and PBDT3-DTBF, by introducing polar oligo (ethylene glycol) (OEG) side chains of different lengths in the donor unit benzodithiophene (BDT) as side chain modification. Studies on solubility, optics, electrochemical, photovoltaic and electrochromic properties are conducted, and the influence of the introduction of OEG side chains on its basic properties is also discussed. Studies on solubility and electrochromic properties show unusual trends that need further research. However, since PBDT-DTBF-class polymers and acceptor IT-4F failed to form proper morphology under the low-boiling point solvent THF solvent processing, the photovoltaic performance of prepared devices is not ideal. However, films with THF as processing solvent showed relatively desirable electrochromic properties and films cast from THF display higher CE than CB as the solvent. Therefore, this class of polymers has application feasibility for green solvent processing in the OSC and EC fields. The research provides an idea for the design of green solvent-processable polymer solar cell materials in the future and a meaningful exploration of the application of green solvents in the field of electrochromism.

## 1. Introduction

Conjugated polymers have broad application prospects in optoelectronic devices, such as organic solar cells (OSCs) [1], organic field-effect transistors (OFETs) [2], organic electrochemical transistors (OECTs) [3], organic thermoelectrics (OTEs) [4,5], organic light-emitting diodes (OLEDs) [6,7], electrochromic devices (ECDs) [8], charge storage [9] and bioelectronics fields [10]. However, most processing of conjugated polymers can only use halogenated solvents, which greatly affects commercial application of conjugated polymers in various fields. Most rigid conjugated polymers are modified with alkyl side chains on the backbone to ensure solubility. To a step further, oligo (ethylene glycol) (OEG) side chain is gradually obtaining more concern with the polarity to utilize in green polar solvent processing [11]. In addition, OEG also features hydrophilicity, high flexibility and excellent ion conductivity [12], which makes the OEG side chain widely used in the modification of OFET [13] and OECT [14] materials. As for organic thermoelectrics [15], the introduction of ethylene glycol side chains under appropriate conditions is also beneficial. However, in other fields of optoelectronic materials, such as OSC and EC, modification research with OEG has just received attention in recent years.

Most highly efficient OSC systems, due to their good solubility in halogenated solvents, choose halogenated solvents (i.e., chloroform or chlorobenzene) for solution treatment to adequately dissolve polymer donors (PDs) and small molecular acceptors (SMAs). Apparently, these halogenated solvents are not sustainable or compatible with industrial production because they are harmful to humans and the environment [16]. To address this important issue, OSC polymers modified with OEG side chains have earned attention due to the above-mentioned merits to substitute the alkyl counterpart in processing of OSC from green solvent in our former work [17]. Furthermore, the flexibility of the OEG allows the side chains to curl for close π-π stacking of conjugated polymer. The augmentation of charge mobility of polymers may also benefit from the inclination to form π-stacked aggregates, which is reckoned to boost the performance of OSCs [18]. Chen [19] et al. report isoindigo-based polymer electron donor units modified with branched OEG side chains (P-OEG). In contrast with the matched control polymer modified with the counterpart alkyl side chains (P-Alkyl), P-OEG displays, as predicted, a smaller π-π stacking distance and bathochromic shift in absorption spectra. Cui [20] et al. introduced OEG side chains to the dicyanodistyrylbenzene-based non-fullerene acceptors (NIDCS) and enhanced dielectric constant ϵ*_r_* to 5.4. Notably, the NIDCS acceptor modified with two triethylene glycol chains (NIDCS-EO3) shows high *V*_oc_ (1.12 V) in OSC device with PTB7 as the polymer donor. Sun [21] et al. designed and synthesized a series of small molecular acceptors, BT2O, BTO and BT4O, with different lengths of OEG side chains in order to induce the self-assembly of Y6 in non-halogenated paraxylene (PX) solution. BTO with triethylene glycol chains doped with PM6:PM7:Y6 delivers the highest PCE of 17.78%. Therefore, the development of a wider variety of OSC materials based on OEG side chain engineering is particularly feasible for the development of the OSC field.

In the field of EC, the research of green solvent processing of materials is also crucial. On the other hand, side chain modification play an important role in the electrochromic properties of conjugated polymers [22], but the effect of OEG side chain is less reported and not sufficiently revealed. Chen [23] et al. designed and successfully synthesized an electrochromic thieno [3,2-*b*] thiophene-based polymer (PmOTTBTD) modified with OEG side chains taking advantage of high ionic conductivity. PmOTTBTD achieved nearly double contrast (42% vs. 24%), fast oxidation switching time and much higher coloration efficiencies than POTTBTD without OEG side chains modification, which demonstrates the overall elevation in electrochromic properties. Hu [24] et al. synthesized and compared a series of functionalized poly (3,4-ethylenedioxy bithiophene)s (PEDTs) with OEG or alkyl side chains to tune their electrochromic properties. The OEG-modified polymers exhibited predictable preferable electrochromic properties, including a lower average switching time and a higher coloration efficiency in contrast with alkyl-modified polymers. Reynolds [25] et al. designed and synthesized three OEG-incorporated propylenedioxythiophene (ProDOT) copolymer electrohromes with excellent electrochromic properties in aqueous electrolyte. However, currently, there are not many research studies that combine EC and OSC properties in one conjugated polymer, exclusively for OEG side chain functionalized polymers.

BDT units have certain electron-giving capability and are often used to construct D-A conjugated polymers in photoelectric fields [26,27,28,29,30,31]. Wei et al. [32] synthesized a series of BDTTBO polymers with different conjugated side chains for ternary blend solar cells with another conjugated polymer donor PTB7-TH and fullerene acceptor PC_71_BM. The power conversion efficiency (PCE) of the optimized ternary blend device of BDTTBO-BT: PTB7-TH: PC_71_BM was enhanced to 10.4%. Then, in 2020, they synthesized a conjugated polymer BDTTBO and then varied the side chain structure by insertion of sulfur atoms and substitution of chlorine atoms to probe the effect of interaction with IT-4F small molecule in their binary blends [33]. Side-chain engineering of modified polymers based on benzo[c] [1,2,5] thiadiazole (BT) unit has also shown recent advances in EC field. Lin [34] et al. designed and electrosynthesized three BT-based polymers with different alkyl side chain decoration, from which DT6FBT polymer with linear hexyl showing the best optical contrast. He [35] et al. synthesized a copolymer combining EDOT unit and BT unit by Stille coupling reaction, exhibiting high switching current ratio and low threshold voltage. Ming [36] et al. inserted 3,4-dihexoxythiophene as π-spacer between EDOT unit and BT unit and prepared the D-π-A-π-D type conjugated polymer poly (BT-Th-EDOT). The π-spacer modification improved electrochromic performance of polymer PBDT, such as optical contrast of 46% and short switching time as 0.4 s.

In this work, based on the efficient PBDT-DTBF polymer system in the OSC field, we changed the hydrophobic alkane side chain into polar oligo-glycol side chains. Different oligo-glycol (OEG) side chains were introduced in donor benzodithiophene (BDT) unit, and three new D-A conjugated polymers, PBDT1-DTBF, PBDT2-DTBF and PBDT3-DTBF, were synthesized. All three polymers have excellent solubility in the conventional halogenated solvent chlorobenzene (CB), CHCl_3_. The donor unit in PBDT1-DTBF was not modified by OEG side chains, with relatively poor solubility. The solubility of PBDT2-DTBF was satisfying in polar solvent *N*-methylpyrrolidone (NMP), N-N-dimethylformamide (DMF), acetonitrile (MeCN), toluene (Tol) and polar green solvent tetrahydrofuran (THF), 2-Methyltetrahydrofuran (2-Me-THF). However, with the further elongation of the OEG side chain, solubility of PBDT3-DTBF has decreased. We speculate through computational simulations that the elongation of the OEG side chain affects the folding dihedral angle of the polymer backbone, and then adjusts polymer chain stacking, which diminishes the solubility of PBDT3-DTBF. Based on a solubility test, the halogen-free and benzene-free solvent tetrahydrofuran was selected as the processing solvent. Because PBDT-DTBF-class polymers and acceptor IT-4F failed to form a good phase morphology under the low-boiling point solvent THF solvent processing, photovoltaic performance of the prepared devices was not ideal. However, films with THF as processing solvent showed relatively desirable electrochromic properties. Therefore, a series of conjugated polymers processable from green solvent were successfully prepared and their possibility of application in organic solar cells and electrochromic applications was demonstrated. It provides insights into the effects of OEG side chains on conjugated polymers and also options for green solvent-processing conjugated polymer materials.

## 2. Results and Discussion

### 2.1. Design, Synthesis and Molecular Weight of PBDT Polymers

To better understand the effect of side chains on polymer solubility and electrochemical properties, we designed a series of polymers allowed to improve solubility while maintaining electrochemical properties in different solvents to explore more environmentally friendly processing conditions as much as possible. The PBDT polymers were selected as the backbone, under consideration that BDT-BT-based D-A polymers have been well-studied and OEG side chains were chosen to tune electronic structure and solubility. In our former work [17], Monomer 12 (Figure 1): 4,7-bis(5-bromo-4-(2,5,8,11-tetraoxatridecan-13-yl) thiophen-2-yl)-5,6-difluorobenzo[*c*] [1,2,5] thiadiazole (DTBF) with two symmetric linear OEGs side chains was first designed and synthesized. Then Monomer 12 reacted with three BDT monomers bearing different lengths of OEG side chains by Stille coupling reactions to form PBDT1-DTBF, PBDT2-DTBF and PBDT3-DTBF in high yields. In the backbone, 5,6-difluorobenzo[*c*] [1,2,5] thiadiazole (BF) unit operated as acceptor, and the BDT unit operated as donor parts. The synthesis routes are visualized in Figure 1 and Figure 2, and the synthetic details of new monomers, characterization data (^1^H NMR, FTIR and HRMS) are marshaled in the Appendix A.

The PBDT polymers are classical and well-studied donor-acceptor (D-A) type polymers: BDT unit with different OEG side chains as donor and BF unit as acceptor. The introduction of F atoms in BF unit lessens LUMO level, thus improving the stability of polymers in the atmosphere [37], and as the F atom is close to the S atom, weak noncovalent interactions are formed, which are known as noncovalent conformational locks (F···S), which also strengthen the planarity of the polymers’ configuration [38]. Moreover, the substitution of CH_2_ group with oxygen atoms gives the OEG side chain a high polarity so that the introduction of OEG side chain into the conjugated polymer will raise the resulting polymer polarity, thereby improving the permittivity and solubility in polar solvent. The OEG chain was employed as a side chain to functionalize conjugated polymers for efficient processing with polar nonhalogenated solvents by virtue of the polarity of OEG chain. Moreover, with just two lone electron pairs in the oxygen atom, the substitution of the oxygen atoms in alkyl chains avoid the two hydrogen atoms in CH_2_ unit as a steric hindrance, making the rotation of covalent bond in OEG chain more flexible [39]. The resultant polymers are expected to demonstrate good compatibility with the polar electrolyte [40].

In the field of organic photoelectric materials, the molecular weight of polymers plays an important role in fine-tuning the phase morphology and photovoltaic properties of OSC. Based on solubility of the polymer material and the refractive index of sample and eluate, DMF was selected as the liquid phase at 80 °C (polystyrenes were selected as the calibration standard). The number-average molecular weight (*M*_n_), weight-average molecular weight (*M*_w_) and the polymer dispersity (*Ð*) were determined by gel permeation chromatography (GPC). PBDT1-DTBF, PBDT2-DTBF and PBDT3-DTBF possessed high *M*_n_ and reasonable *Ð* (Table 1). We speculate that the OEG side chain on BDT unit in PBDT3 can match the side chain on DTBF unit, and that the larger free volume can facilitate the metal transfer of the Stille reaction and the transition state will be more easily formed. Higher molecular weight usually indicates the electronic better-tuned, smooth morphology and desirable photovoltaic properties of OSCs [41,42,43].

### 2.2. Solubility and Thermal Stability

Solubility of polymer is an important performance index in device preparation and processing. The solubility of PBDT1-DTBF, PBDT2-DTBF and PBDT3-DTBF in conventional solvents (such as *o*-DCB, CHCl_3_, MeCN, Tol) and some non-halogenated green solvents (such as NMP, DMF, PC, THF, 2-Me-THF, CPME, D-limonene and ethanol) was determined. These solvents are commonly used as processing solvents in the OSC field. The tests were at concentration of 10 mg/mL, and results are shown in Table 2. All these three polymers showed favorable solubility in the halogen-containing solvents CB and CHCl_3_. PBDT1-DTBF is a typical D-A conjugated polymer without side-chain modification, with relatively poor solubility in other solvents except for THF. However, solubility of the polymers is greater than 3 mg/mL in DMF. Therefore, the molecular weight of PBDT1-DTBF can be measured by using DMF as an eluting agent. Due to the oligo ethylene glycol side chain (OEG = 2) introduced in PBDT2-DTBF, the solubility is obviously improved. At room temperature, thanks to the significant difference in electronegativity between oxygen atoms and carbon atoms in OEG side chains, PBDT2-DTBF can be dissolved in toluene at ambient temperature, in NMP, DMF, 2-Me-THF under heating, and without precipitation even after cooling. The dipole moment (μ) of C-C non-polar covalent bond is much lower than that of C-O covalent bond (0.74 Debye). It is the larger dipole moment of C-O covalent bond which provides the enhanced polarity of OEG chain. It is reported that the introduction of OEG side chains in conjugated polymer will significantly enhance the polarity of polymer and increase the dielectric constant, thus improving its solubility in the polar solvent [44]. However, with the increase of the OEG side chain length, the solubility of polymer PBDT3-DTBF not only did not further improve, but showed a descending trend. We speculate the reason is that PBDT3-DTBF has a much larger molecular weight as compared to PBDT1-DTBF and PBDT2-DTBF. Moreover, the theoretical calculations suggest that the packing between PBDT3-DTBF is much tighter than that in the other two polymers (Figure 1). Past work has shown that there is positive correlation between the ordered aggregation of conjugated polymers and the repeating units of OEG side chains [45]. The effect of different lengths of OEG side chains on the morphology of polymer devices is complicated and requires more in-depth analysis.

The thermal properties of polymers PBDT1-DTBF, PBDT2-DTBF and PBDT3-DTBF were characterized by the heat-loss curves. As shown in Appendix A, all the polymers exhibited 5% weight loss temperature (T_d5%_) above 300 °C, indicating three polymers can fully maintain a stable performance under the temperature conditions for further processing and application.

### 2.3. Theoretical Analysis

To further understand the effect of lengths of OEG side chains on D-A conjugated polymers, density functional theory (DFT) of model compounds BDT1-DTBF, BDT2-DTBF and BDT3-DTBF at B3LYP/6-31 G (d,p) level was performed by Gaussian 16 [46]. As shown in Figure 1, dimer BDT1-DTBF, BDT2-DTBF and BDT3-DTBF were observed to exhibit similar conformations. When the OEG segment was short (*n* = 2), BDT2-DTBF was not significantly affected compared with PBDT1-DTBF, and the dihedral angle between BDT unit and adjacent thiophene units was close (20.5° and 20.4° for BDT1-DTBF and BDT2-DTBF respectively). When the OEG segment grows (*n* = 3), the dihedral angle between BDT unit and BT units decreases to 18.9°. The smaller dihedral angle indicates that the interchain stacking of PBDT3-DTBF is tighter than the other two polymers, performed as lower solubility, which is consistent with our measured results (the solubility changes in 2-Me-THF in Table 2). Furthermore, in the three polymers, the length change of OEG side chains did not have much effect on HOMO/LUMO energy levels. Since the OEG side chain does not have a strong electro-withdrawing effect, it cannot significantly change the HOMO and LUMO levels of polymer [5,47]. Although the HOMO and LUMO levels are quite different from experiment, their trend of data change is consistent, such that the HOMO and LUMO levels of the three polymers are similar. This bias is caused because the simulation objects are small molecule model compounds, while the actual test objects are polymer films.

### 2.4. Fundamental Optical Properties

The ultraviolet-visible (UV-vis) absorption spectra of PBDT1-DTBF, PBDT2-DTBF and PBDT3-DTBF polymers in the chlorobenzene solution and thin-film states are depicted in Figure 2, respectively. The basic optical properties of PBDT1-DTBF, PBDT2-DTBF and PBDT3-DTBF are summarized in Table 3. From Figure 2, there are no prominent shifts about the wavelength of the absorption spectra between the solution and solid states, and the corresponding maximum absorption peak appears slightly redshifted due to the π-π stacking effect between polymer chains. As shown in Figure 2b and Table 3, the absorption peaks at 393 nm (PBDT1-DTBF, film), 396 nm (PBDT2-DTBF, film) and 397 nm (PBDT3-DTBF, film) were distributed to the π-π* transitions along the conjugated backbone. The absorption peaks at 480 nm (PBDT1-DTBF, film), 469 nm (PBDT2-DTBF, film) and 479 nm (PBDT3-DTBF, film) were attributed to the molecular charge transfer (ICT) between the HOMO and LUMO levels [48]. Based on the absorption spectra, the optical band gap ($E_{g}^{opt}$) of PBDT1-DTBF, PBDT2-DTBF and PBDT3-DTBF was calculated and listed in Table 3. The optical band gaps of PBDT1-DTBF, PBDT2-DTBF and PBDT3-DTBF are successively 2.10, 2.12 and 2.10 eV, meaning they are broadband gap polymers. According to previous reports [18], although the OEG side chain is supposed to have a higher electron donor capacity according to the above analysis, the OEG group did not affect the band gap as significantly as predicted, which was also demonstrated in this study.

### 2.5. Electrochemical Properties

The electrochemical properties of PBDT1-DTBF, PBDT2-DTBF and PBDT3-DTBF were studied with ferrocene (Fc/Fc^+^) as the internal standard (Appendix A). The polymer solution at a concentration of 20 mg/mL was spin-cast onto the indium oxide (ITO) glass through a homogenizer at 1300 rpm. The polymer film was placed at 150 °C under vacuum condition for 8 h to remove the residual solvent. In the three-electrode system, the polymer film spin-cast applied as the working electrode, with 0.01 M Ag/AgNO_3_ non-hydro electrode as the reference electrode and platinum wires as counter electrodes. The electrolyte solution used in the test was 0.1 M tetrabutylammonium hexafluorphosphonate/propylene carbonate solution (TBAPF_6_/PC).

The cyclic voltammetry curves for PBDT1-DTBF, PBDT2-DTBF and PBDT3-DTBF are shown in Figure 3, and the corresponding results are enumerated in Table 4. Within the electrochemical window of 0–1.0 V, the three polymers display three oxidation peaks with no reduction peaks. This demonstrates that the process is an irreversible electrochemical doping process. This usually implies that the overdoping phenomenon and electrochemical side reactions already occur at high voltages during oxidation or reduction. Since OEG is a donor unit, the polymer presents a lower onset oxidation potential (*E*_onset_) during forward scanning with the increase of OEG length. Therefore, the *E*_onset_ of PBDT1-DTBF, PBDT2 and PBDT3-DTBF are 0.34, 0.32 and 0.30 V, respectively, showing a decreasing trend. The matched highest-occupied molecular orbit (HOMO) of these polymers is calculated from the starting oxidation potential corrected of ferrocene, and the lowest unoccupied molecular orbital (LUMO) was derived by HOMO and optical band gap. According to equation *E*_HOMO_ = −(4.80 + *E*_onset_), the HOMO energy levels of PBDT1-DTBF, PBDT2-DTBF and PBDT3-DTBF are −5.14, −5.12, and −5.10 eV, respectively. From the equation *E*_LUMO_ = *E*_HOMO_ + $E_{g}^{opt}$, the LUMO levels are −3.04, −3.00 and −3.00 eV for PBDT1-DTBF, PBDT2 and PBDT3-DTBF, respectively. Although the result is quite different from theoretical analysis, their trend of data change is consistent. This bias is caused because the simulation objects are small molecule model compounds, while the actual test objects are polymer films.

The flat-band potentials of samples were measured using the electrochemical method Mott–Schottky plots (Appendix A). All films showing the negative slopes of the linear plots indicate the typical p-type characteristic semiconductors. In general, the *E*_VB_ of p-type semiconductors is very close to the flat band potential [49,50]. The *E*_VB,PBDT1_, *E*_VB,PBDT2_ and *E*_VB,PBDT3_ could be inferred as −0.12 V, −0.12 V and −0.04 V, respectively. Therefore, based on the formula *E*_CB_ = *E*_VB_ − *E*_g_ [51], the *E*_CB_ of PBDT1-DTBF, PBDT2-DTBF and PBDT3-DTBF are −2.22 V, −2.24 V and −2.14 V, respectively.

### 2.6. Photovoltaic Performance

In order to study the photovoltaic performance of PBDT1-DTBF, PBDT2-DTBF and PBDT3-DTBF, the conventional quintuple device structure was employed: ITO/PEDOT: PSS/active layer/PFN-Br/Al, as shown in Figure 4b. In the active layer, PBDT1-DTBF, PBDT2-DTBF and PBDT3-DTBF were used as the donor material, while IT-4F was used as the acceptor material [52]. IT-4F has a strong absorption capacity and high electron mobility in thin films. Moreover, the frontier molecular orbital (FMO) energy levels of PBDT1-DTBF, PBDT2-DTBF and PBDT3-DTBF closely match the acceptor IT-4F, indicating that there are sufficient Δ HOMO and Δ LUMO to achieve effective exciton dissociation in charging process of the interface transfer, and the level matching diagram is shown in Figure 4a [53]. Where PFN-Br is used as the anode interlayer (AIL), and PEDOT: PSS also serves as the cathode interlayer (CIL). The PV parameters are enumerated in Table 5.

Appendix A displays the Nyquist plots of PBDT1-DTBF, PBDT2-DTBF and PBDT3-DTBF films processed from THF. All three films show similar arc radius, indicating similar interfacial charge transfer, which is related to the uneven phase of films processed from THF. The Nyquist plots of PBDT3 sample consist of a semicircle and a capacitive tail at low frequency, showing a lower slope of the straight line in the low frequencies area. The longer OEG side chains in PBDT3 tend to form more ion-trapping cages while limiting ion transport and Faradaic impedance increased.

Through testing, it was found that the device performance of PBDT-DTBF and IT-4F was not ideal, because the PBDT-DTBF conjugated polymer and acceptor IT-4F failed to form smooth and even morphology under the low boiling point THF solvent processing, so they failed to form the appropriate phase region size, which adversely affected the exciton dissociation, charge generation and transmission. Considering the PBDT-DTBF class polymers are new functional materials that are different from the common polymers in the OPV community, the surface energy of polymers could be a factor to take into consideration when blending or screening ideal materials for OSC devices [33].

### 2.7. Spectroelectrochemistry Properties

The spectroelectrochemical properties of PBDT1-DTBF, PBDT2-DTBF and PBDT3-DTBF were studied with UV-vis-NIR to monitor the change in absorbance under different potential.

As chlorobenzene has the best solubility to polymer, the polymer is dissolved in chlorobenzene (20 mg/mL) and coated on ITO electrode to form a thin film. As shown in Figure 5a–c, absorption spectra of PBDT1-DTBF, PBDT2-DTBF and PBDT3-DTBF films are similar. As the potential gradually increases (0–1 V), PBDT1-DTBF, PBDT2-DTBF and PBDT3-DTBF films become fully oxidized and the film color transitions accordingly from orange (neutral state) to dim yellow-green (oxidized state). According to previous research, the neutral color of most dioxythiophene-benzothiadiazole (BT-Th)-based polymers appears green [54]. However, the neutral color of PDTBF polymers appears orange [55], due to the modification of OEG side chains hindering the formation of conformational locks, resulting in the less coplanar configuration and absorption wave blueshifts of the polymers. As OEG chain does not directly connect to thiophene backbone, PBDT2-DTBF and PBDT3-DTBF exhibit very similar electrochemical and optical behaviors.

As the potential is increased to 0.4 V, a gradual impairment of the π-π* transitions occurs, and charge carrier bands at 644–672 nm in the near-IR can be observed to increase slightly. However, as the potential continues to increase to 1 V, the absorption peak in the visible light region shows a descending trend, which also corresponds with the process of over-oxidation of polymer film.

To further test whether these polymers can be processed with green solvents, the polymer is dissolved in THF solvent spin coating to prepare polymer films because the polymer is more soluble in THF solvent. All the PBDT1-DTBF, PBDT2-DTBF and PBDT3-DTBF polymer film-coated ITO glass was heated at 100 °C under vacuum condition for 8 h to remove the residual solvent. As shown in Figure 5 and Appendix A, although the films are cast from different processing solvents, the spectroelectrochemical properties show little difference. Irresistibly, it is difficult for the naked eye to accurately distinguish the slight color changes between the oxidation state and the neutral state. Colorimetric analysis based on the “Commission Internationale de l’Eclairage” 1976 L*a*b* color standards was adopted to quantificationally describe PBDT1-DTBF, PBDT2-DTBF and PBDT3-DTBF polymers and evaluate the effect of casting solvent on the perceived color of films. The a* and b* values range from 127 to −128, representing red to green and yellow to blue. L* depicts the lightness from 0 to 100 representing black to white. The detailed L*a*b* results are listed in Appendix A [56]. Results in this experiment show that the color change of PBDT1-DTBF, PBDT2-DTBF and PBDT3-DTBF films are similar [57]. Interestingly, the color change for the same polymer film processed from THF is slightly more detectable than chlorobenzene as the solvent (Appendix A).

### 2.8. Electrochromic Properties

Polymer films spin-cast by means of before-mentioned fabrication on the ITO glass were applied to study the electrochromic properties. The chronoamperometric and absorbance measurements were carried out on all the films of PBDT1-DTBF, PBDT2-DTBF and PBDT3-DTBF prepared by halogenated solvent CB and polar green solvent THF with an active area of 2.4 cm^2^. Square-wave potential with 0–0.4 V (Ag/AgNO_3_ in 0.1 M TBAPF6/PC) was applied to the polymer film to obtain the function of transmittance (ΔT) at λ_max_ about time and switching time (response time corresponds to 90% of the complete color switching) was calculated.

As shown in Figure 6, the optical contrast of PBDT1-DTBF, PBDT2-DTBF and PBDT3-DTBF dropped rapidly within 300 s, and the apparent trend of ΔT of PBDT3 being dropped was even more pronounced. However, PBDT3-DTBF film exhibits the fastest coloring time ($T_{c}^{0.9}$ = 3.02 s) processed from chlorobenzene and switching time ($T_{c}^{0.9}$ = 4.61 s, $T_{b}^{0.9}$ = 2.85 s) processed from THF (Table 6). By comparing PBDT1-DTBF, PBDT2-DTBF and PBDT3-DTBF films cast from CB, it was noted that after the introduction of shorter OEG side chains (*n* = 2) onto BDT donor, the optical modulation and coloration efficiency (CE) of PBDT2-DTBF film expressed a large drop in contrast with PBDT1-DTBF, while the transmittance change and coloration efficiency were slightly improved when the length of OEG side chain was prolonged (*n* = 3). We infer that the introduction of a short OEG side chain (*n* = 2) affects the interchain stacking resulting in the decrease of conductivity. When OEG chain increases (*n* = 3), the flexibility of side chain becomes expressed. Consistent with Figure 1, the interchain stacking of PBDT3-DTBF is more compact than the other two polymers, which is conducive to the electron conductivity between the chains. Moreover, previous studies have found that triethylene glycol as side chain modification is of appropriate length of side chains to balance the electrochemical and mechanical properties [58]. Overall, the intermolecular interactions of alkoxy-chain-rich materials are usually stronger than alkyl-chain-rich polymers. Intermolecular interaction between polymers with different alkoxy-chain-rich side chain like OEG requires further research [59,60].

Films processed from THF have better color modulation compared with chlorobenzene, which is mentioned above in the colorimetric analysis, but the cyclical stability of films processed from THF is unsatisfactory (Appendix A). In addition, coloration efficiency of the films cast from CB and THF are calculated and compared. It is obvious that films cast from THF display higher CE than CB as the solvent. Compared with the results of films from chlorobenzene, although the whole performances of THF-cast films slightly decreased, the results are sufficiently promising to suggest the feasibility of using non-halogenated green solvents to substitute toxic halogenated solvents.

## 3. Materials and Methods

### 3.1. Materials

The 4-methylbenzenesulfonyl chloride (99%), 1-ethoxy-2-methoxyethane (98%), 1-ethoxy-2-(2-methoxyethoxy) ethane (98%), benzo [1,2-*b*:4,5-*b*′] dithiophene-4,8-dione (98%), Trimethyltin chloride (98%), Tributyltin chloride (98%), 2-(thiophen-3-yl) ethan-1-ol (98%), *n*-Butyllithium (2.5 M), Potassium tert-butoxide (98%), Tris(*o*-tolyl) phosphine(P(*o*-tol)_3_) (98%) and Tris(dibenzylideneacetone) dipalladium-chloroform adduct (Pb_2_ (dba)_3_·CHCl_3_) (98%) were obtained from Energy Chemistry. All the anhydrous solvents using in the synthetic reactions and device processing, such as chlorobenzene (CB), toluene (Tol), tetrahydrofuran (THF), 2-Methyltetrahydrofuran (2-Me-THF), ethyl acetate (EA), ethanol, *N*-methylpyrrolidone (NMP), *N*,*N*-dimethylformamide (DMF), isopropanol (IPA), methanol, propylene carbonate (PC), acetonitrile (MeCN) and so forth are merchandised by Aldrich Chemical and Energy Chemistry.

### 3.2. Synthesis of Polymers

Synthesis of PBDT1-DTBF: The synthesis routes are visualized in Figure 2, Monomer 6 (115.19 mg, 0.20 mmol), monomer 11 (174.93 mg, 0.20 mmol), catalyst Pb_2_ (dba)_3_·CHCl_3_ (6.21 mg, 6.00 mmol), ligand P (*o*-tol)_3_ (14.61 mg, 0.048 mmol) were added to the reaction bottle under the protection of argon and 3 mL anhydrous toluene into the reaction system. The temperature was increased to 110 °C and reacted for 48 h. After the reaction, the polymer was then successively extracted in petroleum ether, methanol, ethanol, ethyl acetate, and trichloromethane with Soxhlet extractor, then finally 67 mg of reddish-brown polymer with metallic luster was obtained. ^1^H NMR is shown in Appendix A.

Synthesis of PBD2-DTBF: The monomer 7 (150.43 mg, 0.20 mmol), monomer 11 (174.93 mg, 0.20 mmol), catalyst Pb_2_ (dba)_3_·CHCl_3_ (6.21 mg, 6.00 mmol) and ligand P (*o*-tol)_3_ (14.61 mg, 0.05 mmol) were added to the reaction vial under the protection of argon, then 3 mL of anhydrous toluene was injected into the reaction system. The temperature was increased to 110 °C and reacted for 48 h. After the reaction, the polymer was then successively extracted in petroleum ether, methanol, ethanol, ethyl acetate and trichloromethane with Soxhlet extractor, and finally 54 mg of the reddish-brown polymer with metallic luster was obtained. ^1^H NMR is shown in Appendix A.

Synthesis of PBD3-DTBF: The monomer 8 (168.05 mg, 0.20 mmol), monomer11 (0.20 mmol, 174.93 mg), catalyst Pb_2_ (dba)_3_·CHCl_3_ (6.21 mg, 6.00 mmol) and ligand P (*o*-tol)_3_ (14.61 mg, 0.05 mmol) were added to the reaction vial under the protection of argon, and then 3 mL of anhydrous toluene were injected into the reaction system. The temperature was increased to 110 °C and reacted for 48 h. After the reaction, the polymer was then successively extracted in petroleum ether, methanol, ethanol, ethyl acetate and trichloromethane with Soxhlet extractor, and finally 60 mg of the reddish-brown polymer with metallic luster was obtained. ^1^H NMR is shown in Appendix A.

### 3.3. Preparation of Electrochromic Film

PBDT1-DTBF, PBDT2-DTBF and PBDT3-DTBF were dissolved in solvent (20 mg/mL) and stirred to ensure the uniformity of the solution. Indium tin oxide (ITO) glass was cleaned successively with deionized water, toluene, acetone and isopropyl alcohol by ultrasound. Then the polymer solution was spin-cast onto the ITO glass with spin speed at nearly 1300 rpm to form films with 100–130 nm thickness. Subsequently, the electrochromic films are heated at 150 °C under vacuum atmosphere for 8 h to remove the residual solvent.

### 3.4. Fabrication and Characterization of the Organic Solar Cells

To assess the photovoltaic performance of OEG side chain modified PBDT polymers, a series of polymer solar cells were fabricated in an inverted quintuple device arrangement of ITO/PEDOT: PSS/active layer/PFN-Br/Al. The patterned ITO glass was sonicated in acetone and isopropanol and washed for 30 min under a UV ozone cleaning system to eliminate the surface tension of the ITO so that the binding force of the polymer and the substrate can get improved. PEDOT: PSS was applied with a thin layer of at 3000 rpm spin-cast and then dry in air for 15 min at 150 °C. The device was then placed into a glove box. The solvent used to dissolve the active material was THF, with the solvent additive of 1,8-diiodooctane (DIO) (0.5%, *v*/*v*). At a donor concentration of 10 mg/mL and acceptor IT-4F, the PBDT-DTBF-containing polymer and IT-4F mixture (1:1, *w*/*w*) were spin-cast on the PEDOT: PSS layer. Then the 0.5 mg/mL PFN-Br methanol solution was cast on the top of the active layer at 3000 rpm for 30 s, forming films about 10 nm thickness. Finally, Al was deposited on the PFN-Br layer under vacuum condition at 80 nm. The effective device area is 0.037 mm^2^. The prepared device was placed in such conditions (AM 1.5 G, 100 mW/cm^2^ simulated sunlight) for current density-voltage (J-V) curve testing and recorded with Keithley 2601 digital table. Extra quantum efficiency (EQE) measurements of polymer solar cells were measured by Crowntech QTest station 1000AD. All device preparation and characterizations were performed in the glove box.

## 4. Conclusions

In this study, a side chain engineering study of the oligo (ethylene glycol) (OEG) of the classical benzodithiophene-benzothiadiazole (BDT-BT)-based conjugated polymer is carried out, and three novel polymers, PBDT1-DTBF, PBDT2-DTBF and PBDT3-DTBF, are designed and prepared by introducing different lengths of OEG side chains into the donor BDT unit to study the effect on solubility, basic optical properties, photovoltaic properties and electrochemical properties of polymers. Due to the introduction of OEG, the polymer showed good solubility in the green polar non-halogenated solvent THF. PBDT2-DTBF shows good solubility even in 2-Me-THF. As for photovoltaic test, however, the photovoltaic performance of polymer solar cell devices is not ideal, because the PBDT-DTBF conjugated polymer and the receptor IT-4F fail to form good phase morphology under the low boiling point THF solvent processing, and fail to form the appropriate phase region size, which adversely affects the exciton dissociation and charge generation and transmission. As for electrochromic properties, choosing THF as the processing solvent, all three polymers showed relatively desirable results. Therefore, this class of polymers has certain application potential for green solvent processing in the OSC and EC fields. Above all, it is necessary to further probe the influence of OEG side chains on conjugated polymers, regulate the solubility of materials and select the appropriate processing solvent to improve the electrochemical properties of materials.

## Data Availability

The authors will make the raw data supporting the conclusions of this manuscript available to any qualified researcher.

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
