# Peer review of "Effects of Different Lengths of Oligo (Ethylene Glycol) Side Chains on the Electrochromic and Photovoltaic Properties of Benzothiadiazole-Based Donor-Acceptor Conjugated Polymers"

_molecules, 2023, doi:10.3390/molecules28052056_

Round 1
Reviewer 1 Report
In this manuscript, the author synthesized three kinds of polymers containing linear ethylene glycol side chains with different lengths, and explored their photovoltaic and electrochromic properties. In the field of OSCs, the performance of this material is not ideal, but its remarkable electrochromic properties have been explored and discussed. This paper can be accepted after major corrections.
1. The grammar should be revised through the whole paper.
2. In line 12, 23 and 104, it is suggested to change “receptor” to “acceptor” .
3. In line 19 and 40, oligo (ethylene glycol), case issues.
4. The overall English needs to be improved. Please seek guidance from a native English speaker is possible. ( Especially “the” could be corrected)
5. In line 41 to 43, “obtaining more concern with the hydrophilicity......In addition......also features hydrophilicity”, “hydrophilicity” was repeated.
6. In line 45, “OEG`s modification......”, there are multiple grammatical errors in the article, which require extensive revision.
7. It is suggested to cite some recent work. For instance, conjugated polymers with branched ethylene glycol side chains for organic thermoelectrics. (https://onlinelibrary.wiley.com/doi/10.1002/anie.202214192) . This study illustrates that the introduction of ethylene glycol side chains under appropriate conditions is beneficial.
8. In line 89, what is “PBDTHDODTHBTff” ?
9. In line 24 to 27, 105 to 109, “the photovoltaic performance of the prepared devices is not ideal.” “......demonstrated their potential in organic solar cells......”. Are these two views contradictory?
10. Line 120, “In our former work......”, why not cite literature? Or the work of this?
11. In line 127, revise “detailsof”.
12. Scheme 1 and 2, the ethylene glycol side chain does not contain hydroxyl groups.
13. In line 134, “BDT units” and “BF units”, why they are plural?
14. “The introduction of F atoms in the BF unit not only lessen the HOMO level so that improve the stability of polymers in the atmosphere” . Non-covalent bonding is best described in the theoretical calculations section. F atoms are electron withdrawing groups that can lower the LUMO energy level, the device is stable in air when the LUMO energy level is below -4.0eV. It is suggested to read the recommended literature and revise the content in the article.
15. In line 148, “There is a methylene spacer between......” this statement is inconsistent with the structure of the polymer.
16. In line 150, “the electron-donating ability of OEG side chains is much higher than conventional alkyl side chains......will get decreased for the modification of OEG side chains” . There is insufficient evidence for this view and it does not correspond to the results of your experiment.
17. Line 153, “plays” change to “play”.
18. Line 156 to 157, polystyrenes were selected as the calibration standard.
19. Table 1, the number-average molecular weight of PBDT3-DTBF is much larger than the other two polymers, try to explain this result and relate to the results of characterization.
20. In line 178, revise “oligoglycol”.
21. Line 189, “this is because......”, grammar error.
22. Line 199, figure S11 is the wrong diagram.
23. What is the reason for choosing these solvents when performing solubility tests? It is advisable to briefly explain the reasons for their selection, low cost, safety and common use in industrial production.
24. Figure 1 contains serious errors, the choice of dimer model for theoretical calculations should indicate, and the alkyl chain on thiophene is simplified to methyl.The work in the theoretical calculation section contains errors and it is recommended that the content be reorganised.
25. Figure 2, I suggest the line graph to express the results, in line 230 to 231, explain the significant red-shift in solid state absorption compared to the solution state.
26. Table 4, HOMO/LUMO energy levels do not correspond to the results of theoretical calculations.
27. Table 5, “PSC photovoltaic properties of PBDT-DTBF class polymers based on different processing solvent”, but in practice only THF was used.
28. Line 324, what is “BT-Th”?
29. Line 358, revise “2.4cm2”
30. 1H NMR spectra of polymers have no integration.
Author Response
- The grammar of the article was re-examined
- “receptor” was changed to “acceptor”
- "Oligo" was transformed into "oligo"
- The grammar of the article was re-examined
- The repetition was deleted
- The grammar of the article was re-examined
- Some recent work were cited as suggested
- "BDT-DTBF" means "benzodithiophene-thiophene-difluorobenzo[c] [1,2,5] thiadiazole"
- The materials show fine electrochromic properties, and the surface energy could be a factor to consider for OSC devices in future work.
- Sorry for the carelessness, the citation was added
- The mistake was revised
- The mistake was revised
- The mistake was revised
- The mistake was revised
- The mistake was revised
- The inconsistent sentence was deleted
- This statement was deleted after considering your opinion
- The mistake was revised
- The mistake was revised
- The mistake was revised
- The mistake was revised
- The mistake was revised
- These solvents are commonly used as processing solvents in the OSC field
- Sorry for the error found in Figure 1, but at present, due to the Chinese New Year holiday, the server of theoretical calculations is not accessible. Here we would like to apply for an extension of 3 weeks for supplementary experimental testing.
- Figure 2 was adjusted as line graphs
- Although the result is quite different from theoretical analysis, their trend of data change is consistent. This bias is caused because the simulation objects are small molecule model compounds, while the actual test objects are polymer films
- The mistake was revised
- "BT-Th" means "dioxythiophene-benzothiadiazole" and the explanation was added in the article
- The mistake was revised
- Sorry that the initial data of 1H NMR spectra was lost and we would like to apply for an extension of 3 weeks for supplementary experimental testing.
Reviewer 2 Report
This work, "Effects of different lengths of oligo (ethylene glycol) side chains on the electrochromic and photovoltaic properties of benzothiadiazole-based donor-acceptor conjugated polymers," developed and synthesized three new D-A conjugated polymers, PBDT1-DTBF, PBDT2-DTBF, and PBDT3-DTBF, by adding polar Oligo (ethylene glycol) (OEG) side chains of varying lengths in the donor unit benzodithiophene (BDT). OEG side chains' effects on solubility, optics, electro-chemical, photovoltaic, and electrochromic characteristics are examined. However, the PBDT-DTBF class polymers and receptor IT-4F failed to establish correct morphology under low-boiling point solvent THF solvent processing, resulting in poor photovoltaic performance. On the other hand, THF-processed films had good electrochromic characteristics. Thus, OSC and EC green solvent processes may use this class of polymers. The research suggests designing green solvent-processable polymer solar cell materials and exploring green solvents in electrochromism. The findings are helpful to the OPV/polymer community. Therefore, I would recommend it be published in Molecules after the following minor issues are addressed:
- The supporting information should be consistent in the font, italics, superscripts, and subscripts. In addition, whether lacking of spaces between sentences in the manuscript should be rechecked. (page 2, line 63)
- The resolution of figure 2 should be revised for clarity.
- Since the authors use Fc/Fc+ reference for cyclic voltammetry measurements, the cycle of Fc/Fc+ should be included for clarity.
- In Scheme 1, those monomers should be revised to compound. Only monomers 6, 7, 8, and 12 are monomers in these three synthesized polymers.
- In Scheme 2, there are two PBDT2-DTBF.
- On page 5, line 191, the authors claimed that "strong aggregation behavior resulting from the rigid polymer backbone. "However, in the three polymers, their backbones are the same; the differences are the side chains.
- To synthesize compounds 1 and 2, did the authors purify the crude before NMR measurement? In addition, the yield should be rechecked since they are the same.
- In the synthesis of compound 3, "methyl partoluene" should be revised.
- For the introduction, several works of side chain engineered polymer at BDT unit under similar backbone structures, and side chain engineered electrochromic polymers should be cited — [Organic Electronics 2019, 71, 185. (doi.org/10.1016/j.orgel.2019.05.002); ACS Appl. Polym. Mater. 2020, 2, 2, 636–646 (doi.org/10.1021/acsapm.9b00998); Polymers 2022, 14(22), 4965 (doi.org/10.3390/polym14224965); Molecules 2022, 27(23), 8424 (doi.org/10.3390/molecules27238424); Chemical Engineering Journal 2020, 390, 124572 (doi.org/10.1016/j.cej.2020.124572).]
- Since the synthesized polymers are a new type of functional material that is different from the common polymers in the OPV community, for the suggestion to the authors' next/future work, the surface energy of polymers could be an issue when blending or screening ideal materials for OSC devices. (See ACS Appl. Polym. Mater. 2020, 2, 2, 636–646.)
- Overall, the intermolecular interactions of alkoxy-chain-rich materials are usually stronger than alkyl-chain-rich polymers. The comparison of intermolecular interaction between these three polymers would be challenging since they are all alkoxy-chain-rich materials. (J. Mater. Chem. A, 2021, 9, 7481; Chem. Mater., 2022, 34, 2059) However, the solubility tests and trends are interesting and provide new findings.
Author Response
- The format was re-examined
- Figure 2 was adjusted
- The reference measurement was added in supporting information
- Mistakes in Scheme 1 and Scheme 2 were corrected
- Unreasonable statements are deleted after considering your opinion
- The purification step is supplemented in the SI and our carelessness about the yield was modified
- The mistake was revised
- Some advances concerning the backbone structure are cited in the introduction as suggested
- Thank you for your advice and recommended literature and review were supplemented in this article
Reviewer 3 Report
1. Give more valuable information in the abstract section.
2. The manuscript contains spelling/grammatical errors. So, the language should be polished thoroughly.
3. Source and purity of all chemicals used should be specified in the experimental section. 4. In the field of photocatalysis, the orbital energy of the semiconductor is very important, but in this paper, the author only calculates the Eg by the ultraviolet diffuse reflection. And it is recommended to supplement the orbital energy of the conduction band and valence band;5. As to the BDT-BT-based D-A polymers, some updated refs could be highlighted and documented, such as Polym. Chem., 2022, 13, 2351–2361; Chem. Commun., 2022, 58, 6653–6656; Org. Chem. Front., 2020,7, 3515-3520; New J. Chem., 2020, 44, 16265-16268; J. Org. Chem. 2019, 84, 14627−14635 and Org. Chem. Front., 2021, 8, 4554–4559
6. there is no EIS and other tests further explain the photovoltaic properties7. Give a part for comparison on this photovoltaic properties by other copolymers.
Author Response
- We thought studies on solubility and electrochromic properties show unusual trends and supplemented the abstract
- The grammar and format were re-examined as suggested
- Our carelessness about the purity of chemicals was revised
- Some advances in the backbone of polymer about BDT and BT were cited in the introduction
- Thank you for your suggestion about orbital energy and EIS. But due to the Chinese New Year, the server of theoretical calculation is not accessible and we would like to apply for an extension of 3 weeks to carry out new experiments and submit the revised version.
Round 2
Reviewer 1 Report
no comments
Author Response
Errors in the theoretical calculations have been corrected and the EIS test was supplemented.
Reviewer 3 Report
my comments 4 and 5 is still not replied, it should be done.
Author Response
The EIS test and the orbital energy of the conduction band and valence band
were supplemented and suggested references were documented.